# The Pathophysiology and Treatment of Pyoderma Gangrenosum—Current Options and New Perspectives

**DOI:** 10.3390/ijms25042440

**Published:** 2024-02-19

**Authors:** Magdalena Łyko, Anna Ryguła, Michał Kowalski, Julia Karska, Alina Jankowska-Konsur

**Affiliations:** 1Department of Dermatology, Venereology and Allergology, Wroclaw Medical University, 50-368 Wroclaw, Poland; alina.jankowska-konsur@umw.edu.pl; 2Student Research Group of Experimental Dermatology, Department of Dermatology, Venereology and Allergology, Wroclaw Medical University, 50-368 Wroclaw, Poland; anna.rygula00@gmail.com (A.R.); michal.kowalski.0597@gmail.com (M.K.); julia.karska@student.umw.edu.pl (J.K.); 3Department of Psychiatry, Wroclaw Medical University, 50-367 Wroclaw, Poland

**Keywords:** pyoderma gangrenosum, treatment, immunosuppression, pathophysiology, biologic drugs, new therapies

## Abstract

Pyoderma gangrenosum (PG) is an uncommon inflammatory dermatological disorder characterized by painful ulcers that quickly spread peripherally. The pathophysiology of PG is not fully understood; however, it is most commonly considered a disease in the spectrum of neutrophilic dermatoses. The treatment of PG remains challenging due to the lack of generally accepted therapeutic guidelines. Existing therapeutic methods focus on limiting inflammation through the use of immunosuppressive and immunomodulatory therapies. Recently, several reports have indicated the successful use of biologic drugs and small molecules administered for coexisting diseases, resulting in ulcer healing. In this review, we summarize the discoveries regarding the pathophysiology of PG and present treatment options to raise awareness and improve the management of this rare entity.

## 1. Introduction

Pyoderma gangrenosum (PG) is a rare neutrophilic dermatosis characterized by a painful, aseptic ulcer. Typically, the disease begins with a papule, pustule, blister, or nodule following trauma, which then rapidly progresses to painful ulceration with a characteristic violaceous border (Figure 1) [1,2,3]. The most common localizations of PG are the lower extremities. However, PG could also appear in the regions around a stoma, postoperative wounds, trunk, face, or upper extremities, depending on the disease subtype (Figure 1) [4,5]. In over fifty percent of cases, PG is accompanied by systemic disease [6].

Six clinical variants of PG can be distinguished: ulcerative, bullous, pustular, vegetative, peristomal, and postoperative [1,5,7,8]. The most frequent is ulcerative PG, also referred to as classical PG, described as an inflammatory pustule or nodule that transforms into a necrotic ulcer, usually occurring on the lower limbs. Bullous PG presents fast-growing, painful bulla on the face or upper limbs that transform into erosion or ulcer. A pustule on the leg or trunk is typical for pustular PG. Vegetative PG manifests with a specific pattern of lesions on the trunk, typically a single ulcer that is less painful, more superficial, and progresses more slowly with a good response to therapy. A papule located directly by the stoma transforming into an ulcer represents peristomal PG. After the surgery on the abdomen or breast, postoperative PG can occur in the form of erythema and unhealing wounds that unite, forming the ulcer.

The specific epidemiology of PG is difficult to estimate since only a few cross-sectional studies have been conducted so far. In a recent American study, the prevalence of PG was evaluated at 58 individuals with PG per 1 million adults [9]. The United Kingdom research indicated an incidence rate of 0.63 per 100,000 person years [10]. PG occurs mostly around the age of 50, with a higher prevalence and incidence among women than men [9,11,12]. It has been reported that the mortality of patients with PG could be threefold higher than in the control group after adjustments for sex and age [10]. There are also data indicating the negative influence of PG on quality of life [13].

The proposed pathophysiology of PG is associated with the interplay between innate and adaptive immunity and the state of autoinflammation with the crucial role of neutrophils [6,14,15,16,17,18,19]. Indicators of the autoimmune character of PG are its correlations with other immune-mediated disorders. Data indicate that over fifty percent of individuals with PG suffer from other autoimmune diseases such as inflammatory bowel disease, rheumatoid arthritis, or hematological malignancies [10]. Pathergy, which is the induction or exacerbation of skin lesions due to trauma, is one of the described phenomena [20]. Certain aspects of PG development are still to be fully discovered.

Diagnosing PG may be demanding, especially when distinguishing between PG and skin infections, and vascular or malignant lesions [21,22,23]. A novel diagnostic tool for PG, the PARACELSUS score is based on the prevalence of the characteristics in the population with PG and consists of major, minor, and additional criteria, each awarded with points differently. Criteria A score of at least 10 points corresponds to a high probability of suffering from PG [2]. Other diagnostic criteria are international PG diagnostic criteria with the use of the Delphi method. They contain four sections—histology, history, clinical examination, and response to therapy. A score of at least 5 points suggests the diagnosis of PG [1]. It is worth noting that results from diagnostic criteria have to be considered with the clinical data.

The treatment of PG is arduous; however, new therapeutic strategies are continuously evolving. The basis of the new therapies derives from the current knowledge of PG molecular pathophysiology. In this literature review, we focus on the latest discoveries regarding the pathophysiology and summarize the treatment of PG to raise awareness and improve the management of this rare disease.

## 2. Pathophysiology

Pathogenesis of PG remains unclear. However, based on the existing research, it can be affirmed that it is a complex and multifactorial process (Figure 2). Possible aberrant origins of inflammation encompass neutrophils, T cells, inflammasomes, keratinocyte apoptosis, and modifications in epigenetic patterns. PG is most commonly considered a disease in the spectrum of neutrophilic dermatoses (ND) [14]. ND is a group of skin disorders characterized by sterile lesions such as papules, pustules, plaques, or nodules caused by neutrophilic infiltration [24]. This might be explained by numerous mechanisms as neutrophil chemotaxis is one of the final stages of the inflammatory cascade [14,17].

Based on the existing reports, it can be inferred that genetic predisposition plays a significant role in the pathogenesis of PG. An animal model with the pathogenic gene of tyrosine-protein phosphatase non-receptor type 6 (PTPN6) and the diminished activity of its protein presented autoinflammatory, neutrophilic skin lesions similar to PG. PTPN6 is known to modulate signals from tyrosine-phosphorylated cell membranes and T cell receptors [25,26,27]. The results were supported by a study with human participants [28]. Another gene with its mutation correlated with PG is considered to be the proline–serine–threonine phosphatase interacting protein 1 gene (PSTPIP1) [28,29]. The pathogenic variant may lead to PSTPIP1-pyrin binding of higher affinity and therefore to the activation of the inflammasome [30]. Inflammasome activates interleukin 1β (IL-1β) that indirectly recruits neutrophils, causing an autoinflammatory state [16,31,32,33]. Data indicate an association between this pathway and pyogenic arthritis, PG, and acne syndrome (PAPA) [34].

PAPA spectrum disorders, such as PASH (PG, acne and hidradenitis suppurativa), PAPASH (PASH and pyogenic sterile arthritis), PsAPASH (PASH and psoriatic arthritis), PASS (PG, acne, and ankylosing spondylitis, with or without hidradenitis suppurativa), and PAC (PG, acne and ulcerative colitis), are associated with genes or chromosomal alterations [35].

Growing evidence suggests that individuals with a genetic predisposition and abnormal activation of the innate immune system create a conducive environment for the development of PG. Genetic defects play a crucial role in disrupting molecular pathways in the discussed diseases. Different molecular pathways are altered depending on the severity of PG [36,37].

Immune system dysregulation plays a crucial role in PG pathophysiology. Based on the abnormalities observed in lesional skin, the role of the inflammatory mediators IL-1β, IL-8, IL-17, and TNF-α has been previously delineated. In PG lesions, abnormal neutrophils and T-cells have been identified.

PG may be induced or exacerbated by trauma (pathergy phenomenon). Physical injury can cause the release of IL-36 and autoantigens from damaged keratinocytes and upregulate the expression of IL-8 and IL-17. The aforementioned mechanism in the presence of predisposing factors may induce PG [15,38,39,40].

PG syndromes suggest the pathophysiological pathways of this disease. The occurrence of PG together with the autoinflammatory disorder—hidradenitis suppurativa, pyogenic arthritis, PG, acne, and hidradenitis suppurativa syndrome (PAPASH) and PG, acne, and hidradenitis suppurativa syndrome (PASH)—can be an indicator of autoinflammation in PG [41,42,43,44].

Although IL-1α and IL-1β are acknowledged as one of the main inflammatory cytokines, only the first cytokine contributes to the development of PG. A missense mutation of the *Ptpn6* gene in mice led to the stimulation of IL-1α pathways, including the receptor-interacting serine/threonine-protein kinase 1 (RIPK1) that caused chronic neutrophilic suppurative inflammation similar to PG [45,46,47]. The study also revealed the involvement of mitogen-activated protein kinase 7 (MAP3K7) and MAP3K5, together with caspase recruitment domain-containing protein 9 (CARD9), in this process [45,46,47,48].

Cytokines suspected of contributing to PG are IL-36A and IL-36G, which are produced by epithelial cells when inflammation occurs. They are known for their role in many autoimmune diseases, including psoriasis, hidradenitis suppurativa, acute generalized pustulosis, Crohn’s disease, or ulcerative colitis, associated with PG [49,50,51,52,53]. It might be hypothesized that minor trauma leads to the release of RNA by keratinocytes, activation of the innate immune response, and degranulation of neutrophils, as well as the release of IL-36 and activation of this cytokine by IL-36 neutrophil-derived proteases. IL-36 has a pro-inflammatory effect by stimulating neutrophils to increase the expression of pro-inflammatory cytokines and by promoting the differentiation of naive T cells towards the Th1 lineage [49,53].

It has been reported that another cytokine associated with neutrophils and the development of PG may be IL-25, also known as IL-17E [18]. Many engaged in inflammation cells produce IL-25—these include dendritic cells, basophils, eosinophils, T helper 2 cells (Th2), as well as keratinocytes [54]. IL-25 triggers macrophages to the production of chemokines CXCL1, CXCL10, and CCL20, which attract neutrophils [55]. Other neutrophil-recruiting molecules with higher expression in PG are IL-8, CCL3, CCL5, or IL-16 [15,41,56,57,58]. The latter acts indirectly on neutrophils by causing the production of neutrophile-chemoattractive substances [19].

The importance of IL-6, IL-8, IL-17, and IL-23 in PG pathomechanisms was suggested by Rubas et al. [59]. They observed increased serum levels of all studied interleukins in a group of 48 PG patients. However, only IL-6 and IL-8 presented an association with studied parameters (localization of skin lesion and CRP for IL-6, age for IL-8) [59].

Therapy with the use of tumor necrosis factor (TNF) succeeded in treating PG and therefore underlined the possible importance of this cytokine in the pathophysiology of PG. TNF triggers the expression of adhesion molecules on blood vessels, which promotes the migration of neutrophils. The priming of the inflammasome is also enabled by the binding of TNF to its receptor, which leads to the characteristic of PG autoinflammation.

The association between PG and adaptive immunity has not been fully established. However, some data that support this theory exist. It has been observed that T cell lymphocytes dominate over other immune cells in the border region of the PG ulcer [60]. T cell clones also occur in this site, suggesting the possible mechanism of response to follicular or dermal antigens in PG development [15,61]. Biopsy of early PG papule revealed an elevated gene expression of T cells attracting chemokines and cytokines, including CXCL9, CXCL10, CXCL11, TNF, interferon gamma (IFNG), IL-17A, IL-8, and IL-36G [15]. T cells accumulated there around vessels and pilosebaceous units [15]. The upregulation of a signal transducer and activator of transcription 1 (*STAT1*) and *STAT4* genes, which induce the production of Th1 transcription factors, and downregulation of the GATA binding protein 3 (*GATA3*) gene, which enhances the expression of the Th2 transcription factor, suggest the promotion of Th1 over Th2 response in the early stages of PG [15].

Another lymphocyte imbalance in PG might be the higher level of Th17-promoting cytokines and the reduction of regulatory lymphocytes T (Treg) [15,60]. IL-23 could have an influence on the predominance of Th17 by triggering pathways including STAT3 and the Janus Kinase 2 (JAK2) [28,61]. Consequently, IL-17A is produced and it recruits neutrophils [62,63]. These processes create the IL-23-IL-17 axis, which may indicate an association between adaptive and innate immunity in the evolution of PG.

Data suggest that PG has a characteristic of an autoimmune disease course of relapses and remissions. Evidence supporting this theory might be the observation of some PG ulcers that disappear without any treatment [62]. The possible explanation of this process could be the presence of anti-inflammatory IL-10 and the upregulation of the forkhead box 3 (FOX3) transcription factor of Treg in the ulcers [15,63]. The healing process of PG should be explored in more detail in future studies to reveal new possible therapeutic strategies.

## 3. Treatment

First, PG is a challenging dermatosis, as there are no national or international guidelines regarding the treatment in this dermatological entity. As a rare disease, there are limited studies considering therapeutic options. To date, only two randomized trials have been conducted [64,65]. Frequently, clinicians are prompted to explore alternative therapeutic approaches for patients who are unresponsive to standard treatments, based on new pathophysiological findings. The primary objective of treatment is to arrest the abnormal inflammatory process, and, typically, this objective is realized through the implementation of immunosuppressive measures. Given that PG often coexists with other autoimmune diseases, reports indicate the effective application of modern therapies in treating accompanying conditions while simultaneously achieving the healing of PG lesions. In this paper, we conduct a review of existing therapeutic methods.

### 3.1. Wound Management

Wound care is an integral part of the treatment [66]. The main principles involve proper cleansing with sterile saline or antiseptic and dressing changes. The dressing should be nonadherent to the wound bed and promote a moist environment that is not overly dry or wet. Moreover, it should be easy to remove to prevent trauma and subsequent pain. Surgical procedures and surgical wound management should be limited to avoid pathergy [67]. Gentle mechanical and autolytic debridement is recommended. Compression therapy reduces localized inflammation and tissue swelling, resulting in increased blood circulation, both directly and indirectly, by promoting better mobility [68].

Negative pressure wound therapy (NPWT) is a method of wound management in which a wound dressing is attached to a vacuum suction machine, applying controlled negative pressure that results in better healing. The role of NPWT is still controversial due to the pathergy effect; however, an increasing number of studies prove its effectiveness. Almeida et al. [69] conducted a systematic review in which improvement was observed in 85.1% of the patients treated with this method. Numerous case reports demonstrated that NPWT has resulted in rapid healing and satisfactory clinical outcomes [70,71,72]. On the other hand, some authors indicate that NPWT alone, without skin grafting, does not accelerate healing time [73].

### 3.2. Topical Therapies

For patients with mild PG, topical therapy can serve as a first-line treatment option. The data regarding this therapeutic approach are, in fact, limited. The most commonly used topical therapy is clobetasol propionate twice daily, but other high-potency or superpotent topical corticosteroids can be administered. Besides corticosteroids, topical calcineurin inhibitors are also described in the literature as a treatment regimen. Thomas et al. [74] conducted a prospective cohort study aimed at estimating the effectiveness of topical therapies (topical corticosteroids [classes I–III] and tacrolimus 0.03% or 0.1%) in the treatment of PG. The study involved 66 PG patients, with 49 treated with clobetasol propionate 0.05%, 10 with tacrolimus 0.03% or 0.1%, and 8 using other topical preparations. In the clobetasol group, twenty patients (43%) benefited from the treatment. However, due to comorbidities, nine of them received additional systemic therapy (azathioprine, TNFα inhibitor, and tetracyclines) that might have influenced the treatment outcomes. The median healing time was 145 days. In the topical tacrolimus group, resolution of lesions was observed in half of the group [74]. The comparison between the effectiveness of studied topical regimens was not analyzed as the study was not randomized. Another study compared the effectiveness of topical corticosteroids and topical tacrolimus in the management of peristomal PG. In total, 13 patients received topical clobetasol 0.05%, and 11 received topical tacrolimus 0.3%. Topical tacrolimus showed significantly better effectiveness in managing peristomal PG than corticosteroids. Moreover, the studied calcineurin inhibitor was more effective in the management of lesions larger than 2 cm in diameter [75].

### 3.3. Intralesional Therapies

In the literature, reports indicate the effectiveness of intralesional drug administration in pyoderma gangrenosum. However, this approach is controversial due to the potential risk of pathergy. The implemented therapy in such cases may result in an adverse outcome by inducing the formation of new ulcers or enlarging existing ones. Nevertheless, existing case reports describe the successful administration of intralesional corticosteroids, methotrexate, and cyclosporine A [76,77]. The latest report described a successful intralesional application of infliximab in a patient with PG and systemic lupus erythematosus (SLE) [78].

### 3.4. Systemic Therapies

In the case of extensive, rapidly progressing, or treatment-resistant skin lesions, systemic treatment is recommended. The therapy usually begins with the use of fast-acting immunosuppressive agents, primarily systemic steroids, or CyA, but other immunosuppressive and immunomodulatory drugs are also used in everyday practice.

#### 3.4.1. Corticosteroids

Systemic corticosteroids are the most common first-line treatment option, as the advantage of the mentioned therapy is, in most cases, rapid response [79]. The dose of prednisone ranges from 0.5 to 2 mg/kg per day [80]. Usually, the initial dose starts from 40 to 80 mg daily. Higher doses can be administered in case of rapidly progressing ulcers. In times when disease activity is controlled, a gradual reduction of the dosage is recommended [77,81]. In severe cases, intravenous pulse corticosteroids (0.5–1 g methylprednisolone per day for one to five days) can be administered [82,83].

Despite the frequent usage of corticosteroids in clinical practice, the majority of data regarding their efficacy are derived from case reports and nonrandomized studies. However, a STOP GAP study, designed to determine whether CsA is superior to prednisolone for the treatment of pyoderma gangrenosum, was one of the randomized controlled trials that provided high-quality research outcomes. A total of 53 patients were included in the prednisolone group. Patients were treated with a prednisolone dose of 0.75 mg/kg/d (maximum 75 mg/d). Investigators analyzed the speed of healing over six weeks, time of healing, global treatment response, resolution of inflammation, self-reported pain, quality of life, number of treatment failures, adverse reactions, and time to recurrence. The conclusion indicated that prednisolone and CsA do not differ across reported outcomes. By six months, ulcers had healed in 47% of patients in the prednisolone group. Furthermore, wound healing progressed within a comparable timeframe in both groups. In the prednisolone group, 66% of patients suffered from adverse effects, most commonly infections requiring hospital admission or parenteral antibiotics [84].

Prolonged corticosteroid therapy is associated with several side effects, such as infections, hyperglycemia, osteoporosis, hypertension, and others, which is why steroid-sparing agents are added to the therapy. Typically CsA, MMF, or dapsone are included as a second agent to maintain an improvement as the steroid is withdrawn [85]. In patients who do not respond to other therapeutic options and require long-term corticosteroid therapy, deflazacort may be the best option.

#### 3.4.2. Cyclosporine A (CsA)

As mentioned above, CsA is considered one of the initial treatments in patients with PG. The recommended initial dose of CsA is 2.5 to 5 mg/kg/d, which can be subsequently tapered as tolerated [86]. The effectiveness of CsA was also presented in several case reports where patients were successfully treated with CsA [87,88,89].

During the discussion on corticosteroid use in PG, reference was made to the STOP GAP study. Within the group receiving cyclosporine, a total of 59 patients were included. Patients were administered a dosage of 4 mg/kg/day (maximal dose 400 mg/d). As mentioned before, CsA and prednisolone seem to have similar effectiveness. The results of this study demonstrated that, by six months, ulcers had healed in 47% of participants in the CsA group [84]. CsA side effects were noted in 68% of patients, and the most common was renal toxicity. This study showed that the efficacy of both prednisolone and CsA is similar in PG patients, but less than 50% of them noticed healing of lesions by six months in both groups. Moreover, approximately one-third of patients in both treatment groups experienced a recurrence after a median duration of 582 days. Based on all the above, when selecting a therapeutic option, it is essential to consider comorbidities and patient preferences.

#### 3.4.3. Other Immunosuppressants

There are reports of using conventional immunosuppressants such as mycophenolate mofetil, methotrexate, and azathioprine in the PG treatment.

##### Mycophenolate Mofetil (MMF)

Mycophenolate mofetil (MMF) demonstrated satisfactory therapeutic effects in patients with PG. Most data are derived from retrospective studies. This medication is typically administered in conjunction with other therapeutics. Eaton et al. [90] described a series of 7 cases treated with MMF. The dosage regimen started from 0.5 mg daily to 1 mg twice daily. The maximal dose ranged from 0.5 mg 4 times a day to 2 mg twice daily. In 6 out of 7 patients, they observed a reduction in ulcer size, and in 4, they observed complete healing. The onset of response ranged from 4 to 12 weeks. In two patients, MMF was the only systemic treatment. Among the reported adverse effects, anemia has been documented in one patient [90].

Another retrospective study was conducted by Li et al. [91] to examine the efficacy and safety of MMF in PG patients. In 11 (42.3%) patients, MMF was the first steroid-sparing agent. The initial dose ranged from 1 g to 2 g daily, reaching 2 g or 3 g daily. In total, 22 out of 26 patients benefited from the therapy. The mean time of treatment was 12.1 months. Researchers categorized patients into excellent, good, and no-clear response groups. In excellent responders, the average time to heal was 9.71 months, while in good responders it was 30.8 months. More than half of the patients experienced adverse effects, most often gastrointestinal (*n* = 6) and hematologic (*n* = 3) adverse effects, and infections (*n* = 3) [91].

Most recent report of MMF use in PG as a adjunctive therapy was reported by Hrin et al. [92]. In the 10-year retrospective study, they identified 14 patients treated with MMF and prednisone. The initial and maximal dosage of MMF ranged from 1 g to 2.5 g daily. Improvement of skin lesions was observed in 93% within 12 months. Total healing was noticed in 5 patients, while significant improvement was reported in 4 of them. Half of the subjects experienced adverse effects such as hematologic suppression (21%), gastrointestinal upset (21%), edema (14%), and shortness of breath (14%) [92].

The above studies suggest that mycophenolate mofetil (MMF) is particularly effective in individuals for whom previously employed preferred treatment methods have proven unsuccessful. The addition of MMF to therapy resulted in an improvement in skin lesions.

##### Methotrexate (MTX)

Data on the efficacy and safety of MTX in the treatment of PG are limited. Recently, Williams et al. [93], in their retrospective study, presented 33 patients treated with MTX. The initial dose ranged from 5 mg to 20 mg weekly. Most patients (97%) received concomitant prednisone. Complete response after 4 months was observed in 6% of patients, while partial response was noted in 52% of patients. Side effects were observed in 21% of patients, such as mild gastrointestinal upset, infection, mouth ulcers, and fatigue. This study suggests that MTX may be used in PG patients and has the potential to decrease the corticosteroid dose in the treatment of PG. Moreover, MTX is well tolerated by patients and is characterized by a low incidence of adverse effects [93].

##### Azathioprine

The use of azathioprine in the treatment of PG has been reported. Similarly to MTX, the data regarding this treatment regimen are ambiguous. The majority of them originate from case reports [94,95]. Azathioprine is particularly effective in patients with coexisting conditions such as IBD, rheumatoid arthritis, and other autoimmune disorders, for which this medication is employed as a standard therapeutic approach.

#### 3.4.4. Immunosuppressive Antibiotics

Several antibiotics, besides their antimicrobial properties, present anti-inflammatory effects. The representatives, i.e., dapsone and minocycline, are successfully used in neutrophil-mediated diseases [96].

##### Dapsone

The literature regarding PG treatment with dapsone is sparse and includes several case reports and a retrospective analysis of 27 patients treated with dapsone [97,98,99,100]. It is hypothesized that the effectiveness of this drug in neutrophilic dermatoses is related to its ability to inhibit neutrophil adherence to antibodies. In published studies, dapsone is commonly used as an adjuvant and steroid-sparing agent in doses ranging from 50 to 150–200 mg daily. In their retrospective study, Din et al. [100] analyzed 27 individuals treated with dapsone. In their population, patients received dapsone in combination with other therapies. The study demonstrated a 96.9% response rate with complete healing noted in 15.6% and a partial improvement seen in 81.3% of the patients. The average time to respond was 5.3 weeks. Approximately one-third of the patients experienced adverse events, with hemolytic anemia being the most common [100]. Screening for glucose-6-phosphate dehydrogenase (G6PD) deficiency should precede dapsone therapy since it increases the risk for hematologic toxicity. This therapeutic option works especially well in people who have small-diameter and superficial lesions.

##### Minocycline

Minocycline is a drug that inhibits protein synthesis by binding ribosomal subunits of bacteria and can be used in the treatment of PG. In the available literature, minocycline is usually administered at a dose of 100 mg twice daily and used in combination with other therapeutics, such as oral prednisolone and sulfasalazine [101,102]. In one case, minocycline caused adverse effects such as hyperpigmentation [103].

#### 3.4.5. Intravenous Immunoglobulin (IVIG)

The favorable safety profile of intravenous immunoglobulin (IVIG) renders it an attractive therapeutic alternative for individuals with severe PG who cannot endure the adverse effects of conventional immunosuppressive agents or are already significantly immunocompromised.

In a systematic review, Song et al. [104] summarized data from cases and case series. A total of 49 patients were included of which 43 had complete or partial response. Complete healing was reported in over half of the population (*n* = 26). It should be mentioned that in 43 patients systemic corticosteroids were administered regardless of IVIG treatment. Nearly three-quarters of the patients were treated with 2 g/kg or higher dosage. The average time to initial response was 3.5 weeks and the duration of the treatment was 5.9 months. Authors indicated that the time to initial response was dose dependent—the higher the dose, the shorter the time to response. There were some adverse effects reported, such as nausea (12%) and headache (4%) [104].

A summary of discussed systemic therapies is presented in Table 1.

#### 3.4.6. Tumor Necrosis Factor-α (TNF-α) Inhibitors

##### Infliximab

Infliximab is a chimeric mouse/human monoclonal IgG1 antibody against TNF-α. It prevents the binding of TNF-α to its receptors by interacting with both the soluble and transmembrane forms of the cytokine [105].

Regarding TNF-α inhibitors in PG therapy, the most scientific data concern the use of infliximab. It is warranted in the context of coexisting IBD, as infliximab is recommended in the treatment of the mentioned medical condition. A randomized placebo-controlled trial assessing the efficacy of infliximab in 30 patients with PG and IBD was conducted. After randomization at week 0, one group (*n* = 13) received an infusion of infliximab, and the second group (*n* = 17) received a placebo. At week 2, significantly more patients in the infliximab group observed improvement (46%) compared with the placebo group (6%). Patients from the placebo group, without a response, were offered infliximab. Overall, 29 patients received infliximab, with 69% demonstrating adequate results. The remission rate at week 6 was 21% [64]. In PG, infliximab is administered in standard doses, i.e., 5 mg/kg at weeks 0, 2, and 6, followed by infusions every 6 to 8 weeks.

Several retrospective analyses were performed in patients with PG receiving infliximab. A retrospective study of 13 patients with IBD and PG treated with infliximab demonstrated complete healing of the skin lesions. The mean time to respond was 11 days and the mean time to complete healing was 86.1 days. All patients received standard doses of infliximab [106]. Another retrospective observational study presented the data of 67 patients treated due to PG. Infliximab was used in 24 patients and, in 22 of them, it contributed to complete remission [107].

The above are just the largest reported studies regarding infliximab use in PG. Other studies are presented in Table 2.

Adverse effects of infliximab treatment include infusion reactions, infections, demyelinating disease, and heart failure [108].

**Table 2 ijms-25-02440-t002:** A summary of clinical studies concerning infliximab.

Authors (Year)	Biologic Drug	Dosage Regimen	Study Type	Comorbidities	Efficacy
T.N. Brooklyn et al. (2006) [64]	Infliximab	5 mg/kg i.v. at weeks 0, 2, and 6, followed by infusions every 6 to 8 weeks or placebo at week 0 with possible switch at week 2	Randomized placebo- controlled trial	CD (*n* = 12) UC (*n* = 6) no IBD (*n* = 11)	Out of 29 patients in infliximab group, 20 (67%) demonstrated adequate response
M. Regueiro et al. (2003) [106]	Infliximab	5 mg/kg i.v.	Multicenter retrospective study	IBD in all cases	Complete healing in all 13 cases
F. Argüelles-Arias et al. (2013) [107]	Infliximab	5 mg/kg i.v.	Retrospective observational study	IBD in all cases	Out of 24 patients, 22 (92%) demonstrated complete healing
T. Ljung et al. (2002) [109]	Infliximab	5 mg/kg i.v.	Case series (*n* = 8)	CD	Complete healing in 3 (37%) cases, partial healing in 3 (37%)
F. Salehzadeh et al. (2019) [110]	Infliximab	100 mg i.v.	Case report	none known	Full recovery in 2-year period
M. R. Kaur et al. (2005) [111]	Infliximab	3 mg/kg i.v.	Case report	none known	Full recovery in 4-month period
L. Đ. Betetto et al. (2022) [112]	Infliximab	10 mg/kg i.v.	Case report	UC	Satisfactory result

i.v.—intravenously; CD—Crohn’s disease; UC—ulcerative colitis; IBD—inflammatory bowel disease; *n*—number.

##### Adalimumab

Adalimumab is a fully human monoclonal antibody against TNF-α. The use of adalimumab is supported by a 52-week, phase 3 open-label study of 22 patients with PG. Patients received adalimumab during a 26-week treatment period and another 26-week extension period. The dosage regimen was 160 mg at week 0, 80 mg at week 2, and 40 mg every week from week 4. At week 26, 12 of 22 patients (54.5%) reached a satisfactory outcome. In this study, an adverse effect, which was an infection in one patient, was observed. Based on the results of this trial, we can assume that adalimumab is effective in PG treatment [113]. Moreover, in a retrospective observational study, the healing of PG lesions after adalimumab was observed in 7 IBD patients [107]. There are also a few case reports where the use of adalimumab shows adequate effects [114,115,116]. Nevertheless, some reports suggested that adalimumab may paradoxically provoke PG, which is why this therapeutic method should be administered with caution [117,118,119]. Currently, there is an ongoing observational study of adalimumab for the treatment of pyoderma gangrenosum [120].

A summary of clinical studies concerning adalimumab is presented in Table 3.

##### Etanercept

Even though etanercept seems to be effective in PG treatment, there were no randomized clinical trials conducted. Etanercept is used in a dose of 50 mg once or twice a week, and then 50 mg every other week for 6 months [66]. There are cases described in the literature where etanercept was successfully used [121,122,123,124,125,126,127,128,129]. Ben Abdallah et al. [130], in a semi-systematic review, summarized available data regarding the efficacy of TNF inhibitors in PG and compared the clinical effectiveness of etanercept, adalimumab, and infliximab. Etanercept was used in 36 patients and constituted the smallest subgroup. The response rate was estimated at 83%, while a complete response was noted in half of the patients. In total, 17% of patients did not respond to the treatment. There were no significant differences between the groups; however, etanercept presented less favorable responses [130].

Importantly, as in the case of adalimumab, there is a report of the development of PG in a patient treated with etanercept due to psoriatic arthritis, which indicates an ambiguous role of this drug in PG therapy [131].

A summary of clinical studies concerning adalimumab is presented in Table 4.

#### 3.4.7. Ustekinumab

Ustekinumab is an IL-12/IL-23 antagonist that can be used in PG management. According to the world literature, there are currently 13 cases of successful treatment with ustekinumab reported [118,133,134,135,136,137,138,139,140,141,142,143,144]. The suggested dose of ustekinumab is 90 mg, twice at 4-week intervals, then the same dose every 8 weeks. A summary of clinical studies concerning adalimumab is presented in Table 5.

#### 3.4.8. IL-1 Antagonists

IL-1 antagonists utilized in PG treatment include canakinumab and anakinra [66]. Canakinumab, a human anti-IL- 1b monoclonal antibody was used in a prospective, open-labeled study of 5 patients with PG. Canakinumab was administered in a dose of 150 mg at weeks 0 and 2, and 150–300 mg at week 4 if needed. Four of them were completely healed [145]. Cases of complete resolution of skin lesions after the use of canakinumab have also been reported in the literature [146,147]. Anakinra, a recombinant, non-glycosylated form of IL-1 receptor antagonist used in a dose of 100 mg once a day from 8 weeks to 10 months, was administered only to a few patients described in case reports [148]. A summary of clinical studies concerning adalimumab is presented in Table 6.

#### 3.4.9. IL-17 Inhibitors

IL-17 inhibitors are a group of novel biological drugs that inhibit the activity of IL-17. The available literature describes the effects of treatment with three representatives of this group (secukinumab, ixekizumab, and brodalumab) [149,150,151]. The reports regarding the use of this group of biologic drugs and PG is ambiguous. Secukinumab, a first-in-class fully human monoclonal antibody against interleukin-17A, was successfully used in several case reports [149,152,153]. However, there are existing data on the paradoxical reactions to it. Few authors reported the development of PG in patients treated with secukinumab [141,154,155]. There is one case report on the successful use of brodalumab, another human monoclonal antibody to the interleukin-17A receptor [151]. Interestingly, Sadik et al. [156] reported induction of PG, palmoplantar pustulosis, and sacroiliitis after switching from secukinumab to brodalumab in plaque psoriasis patients. Kao et al. [150,157] documented effective treatment of PG lesions in four patients after ixekizumab therapy. Nonetheless, there is a case report of PG manifestation in a patient undergoing ixekizumab treatment [158]. The majority of reported cases of PG induction are associated with the alteration of biologic drugs in patients with psoriasis. Therefore, transitioning from one medication to another should be approached judiciously. The existing case reports and case series on their successful use of IL-17 inhibitors are presented in Table 7.

##### IL-23 Inhibitors

IL-23 inhibitors, including guselkumab, tildrakizumab, and risankizumab, are a group of new biological drugs targeting the IL-23 pathway, widely used in the treatment of moderate to severe psoriasis and, partially, psoriatic arthritis (to date, only risankizumab and guselkumab are registered for this indication) [159]. Guselkumab is a fully human monoclonal antibody specifically targeting the p19 subunit of IL-23. Risankizumab is a fully human immunoglobulin (Ig)G monoclonal antibody that binds with high affinity to the p19 sub-unit of IL-23. Tildrakizumab is another high-affinity, humanized, IgG1-κ antibody targeting the p19 subunit of IL-23. Despite these representing a novel class of medications, there are several cases reporting successful use of this group of biologics in PG treatment [160,161,162,163,164,165,166,167]. A summary of clinical studies concerning adalimumab is presented in Table 8.

## 4. Future Directions

Undoubtedly, the future of PG treatment lies in targeted therapies. Ongoing research on the pathogenesis of the disease and emerging insights into the pathomechanisms provide hope for the integration of both existing and novel molecules in the treatment paradigm.

### 4.1. Janus Kinase Inhibitors (JAKi)

As in many other dermatological conditions, Janus Kinase inhibitors (JAKi) may be the future of the treatment of PG. Tofacitinib is an oral JAK-1 and JAK-3 inhibitor that has been approved for the treatment of rheumatoid arthritis and ulcerative colitis, plaque psoriasis, atopic dermatitis, vitiligo, and alopecia areata. Cases are reporting successful resolution of PG ulcers in patients treated with JAKi due to concomitant medical conditions. Other JAKi including tofacitinib, baricitinib, upadacitinib, and ruxolitinib were reported as effective treatment options [168,169,170,171]. We present available case reports in Table 9.

Currently, there is an ongoing phase 2 open-label, proof-of-concept, study of baricitinib for the treatment of pyoderma gangrenosum [172] (Table 10).

**Table 9 ijms-25-02440-t009:** A summary of clinical studies concerning Janus Kinase inhibitors.

Authors (Year)	JAKi	Dosage Regimen	Age and Gender	Comorbidities	Efficacy
B. Kochar et al. (2019) [173]	Tofacitinib	(1) 5 mg p.o. twice daily (2) 5 mg p.o. twice daily (3) 5 mg p.o. twice daily, increased to 10 mg twice daily due to not complete healing	(1) 49-year old female (2) 24-year old male (3) 34-year old male	all patients with Crohn’s disease and concomitant arthritis previously resistant to various biologics	(1) Complete healing after 12 weeks (2) Complete healing after 12 weeks (3) Continuous improvement without corticosteroids
P.S. Olavarria et al. (2021) [174]	Tofacitinib	10 mg p.o. twice daily	69-year old female	ulcerative colitis and arthralgias	Complete healing after 4 weeks
L. G. M. Castro (2023) [168]	Baricitinib Tofacitinib	2 mg p.o. twice daily for 39 days 5 mg p.o. twice daily for 120 days	73-year old male 79-year old female	familial Mediterranean fever none known	Complete healing with no relapse
M. R. dos Santos et al. (2023) [169]	Upadacitinib	15 mg p.o. daily	45-year old female	rheumatoid arthritis	Complete regression after 6 weeks
M. Scheinberg et al. (2021) [170]	Baricitinib	4 mg p.o. daily	71-year old female	IgA multiple myeloma in remission	Complete regression after 5 weeks
S. Nasifoglu et al. (2018) [171]	Ruxolitinib	NA	64-year old female	polycythemia vera	Complete healing

p.o.—orally; JAKi—Janus Kinase inhibitors; NA—not applicable.

### 4.2. Spesolimab

An example of medications that may find application in PG includes IL-36 inhibitors. Recently, Guénin et al. [175] reported successful use of spesolimab in two patients with refractory PG. Patients were administered 900 mg of spesolimab intravenously every 4 weeks. During therapy, one patient developed epididymitis without further complications after antibiotic therapy. Another report of successful spesolimab use was presented by Ma et al. [176]. Currently, there is an ongoing clinical study to evaluate the feasibility of spesolimab in PG treatment [177]. Estimated enrollment is approximately 20 participants and the completion of the trial is planned for 09.2025 (Table 10).

### 4.3. Vilobelimab

Vilobelimab, also known as IFX-1, is a complement C5a inhibitor that was tested in phase IIa open-label trial [178]. Nineteen patients were enrolled in the study and were divided into three arms. The first cohort (*n* = 6) received 800 mg of IFX-1 twice weekly for 12 weeks, following an initial phase involving 3 doses of 800 mg on days 1, 4, and 8 of the study, with a subsequent three-month observational period. The second cohort (*n* = 6) received vilobelimab 1600 mg every 2 weeks, with the option to increase the dose from day 57 to 2400 mg every two weeks, and the third cohort (*n* = 7) received 2400 mg every 2 weeks. After the promising results of phase II, phase III randomized, double-blind, placebo-controlled, multicenter trial is currently recruiting patients [179]. Estimated enrollment is approximately 90 participants and the completion of the trial is planned for February 2026 (Table 10).

**Table 10 ijms-25-02440-t010:** A summary of ongoing, currently recruiting registered trials.

Study Number	Medication	Study Phase/Type	Estimated Enrollment	Estimated Study Completion
NCT05964413 [179]	Vilobelimab	3	90	13 February 2026
NCT04750213 [120]	Adalimumab	observational	60	31 August 2025
NCT06092216 [177]	Spesolimab	4	20	September 2025
NCT04901325 [172]	Baricitinib	2	10	5 December 2024

## 5. Conclusions

Treatment of PG is difficult and complex, and the lack of clear recommendations supported by large randomized studies is one of the problems faced by patients with PG. Even though there are currently many therapeutic methods with greater or lesser effectiveness, the key first-line drugs appear to be fast-acting immunosuppressants, such as systemic steroids and cyclosporine to reduce the disease burden, followed by slow-acting immunosuppressive agents with more favorable safety profiles and biologics. However, it is worth emphasizing that targeted therapies seem to be the most promising future option for the effective treatment of PG. Most scientific evidence refers to TNFα inhibitors—infliximab and adalimumab. Numerous case reports support the use of ustekinumab. As in many dermatological conditions, JAKi seems to be a future valuable therapeutic option in PG treatment. Ongoing clinical trials may yield new treatment alternatives, such as spesolimab or vilobelimab.

It is worth noting that rare pathological entities should not be overlooked, and only collaborative efforts across multiple centers can yield credible research results and bring about changes in the treatment of PG.

## Figures and Tables

**Figure 1 ijms-25-02440-f001:**
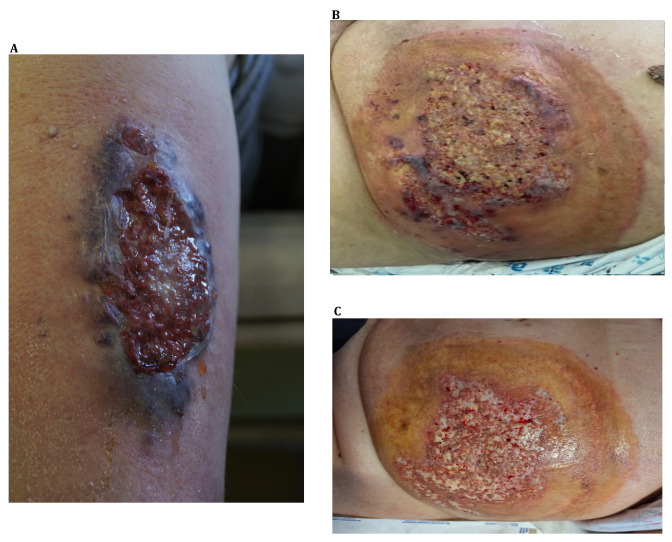
Clinical presentation of pyoderma gangrenosum. (**A**) A purulent ulcer with a raised, violaceous border localized on the lower extremity in the course of PG in a patient with ulcerative colitis. (**B**) Extensive purulent ulceration with a ragged, violaceous border on the abdomen in a patient with acute myeloid leukemia. (**C**) The same lesion after intensive 2-week treatment with cyclosporine A and high doses of prednisone.

**Figure 2 ijms-25-02440-f002:**
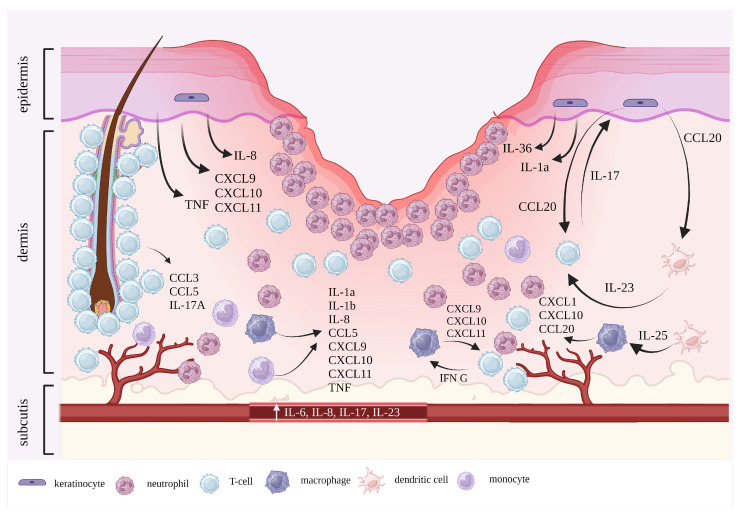
Pathophysiology of pyoderma gangrenosum. The pathophysiological mechanisms underlying the development of PG are complex and involve neutrophils, keratinocytes, T-cells, and other immune cells that produce pro-inflammatory cytokines. The clinically evident undermined border of the ulceration is attributed to the infiltration of neutrophils in the dermis. This figure was created with BioRender.com.

**Table 1 ijms-25-02440-t001:** The summary of systemic therapies.

Authors (Year)	Systemic Drug	Dosage Regimen	Method of Administration	Others
T. Yamauchi et al. (2003) [82]	Methylprednisolone	1 g for 3 days	i.v.	Dosage reduced within 2 weeks—therapy maintained with 30 mg prednisolone daily for 6 months
B. Ambooken et al. (2014) [83]	Dexamethasone	100 mg in 500 mL 5% dextrose infused over 3–4 h on 3 consecutive days	i.v.	9 pulses at 28 days intervals
A. D. Ormerod et al. (2015) [84]	Prednisolone	0.75 mg/kg/day; maximum dose 75 mg/day	p.o.	-
A. D. Ormerod et al. (2015) [84]	Cyclosporine A	4 mg/kg/day; maximum dose 400 mg/day	p.o.	-
P. A. Eaton et al. (2009). [90]	Mycophenolate mofetil	Initial dose 0.5/day or 1g/day; maximal dosages from 0.5 g 4 times daily to 2 g twice daily	p.o.	-
J. Li et al. (2013). [91]	Mycophenolate mofetil	1 g or 2 g total daily	p.o.	The maintenance dose was 2 g or 3 g total daily; the average duration of treatment was 12.1 months
M. L. Hrin et al. (2021) [92]	Mycophenolate mofetil	1g to 2.5 g daily	p.o.	-
J. A.Williams et al. (2023) [93]	Methotrexate	5–25 mg	ND	97% received concominant prednisone
P. Sardar et al. (2011) [94]	Azathioprine	1 mg/kg daily	p.o.	Patient was unresponsive to systemic steroid and dapsone
E. Galun (1986) [97]; R.E. Brown (1993) [98] L. A. Teasley et al. (2007) [99]; R. S. Din et al. (2018) [100]	Dapsone	50–200 mg daily	p.o.	Screening for glucose-6-phosphate dehydrogenase (G6PD) before and during treatment due to hematologic toxicity
P. D. Shenefelt et al. (1990) [101]; N. J. Reynolds et al. (1996) [102]	Minocycline	100 mg twice daily	p.o.	Combination with other therapeutics
H. Song et al. (2018) [104]	Intravenous immunoglobulin	2 g/kg	i.v.	The mean time to initial response of 3–5 weeks

i.v.—intravenous; p.o.—oral; ND—no data.

**Table 3 ijms-25-02440-t003:** A summary of clinical studies concerning adalimumab.

Authors (Year)	Biologic Drug	Dosage Regimen	Study Type	Comorbidities	Efficacy
F. Argüelles-Arias et al. (2013) [107]	Adalimumab	160/80 mg given s.c. at 0 and 2 weeks, and then every 2 weeks	Retrospective observational study	IBD in all cases	7 patients, complete response
K. Yamasaki et al. (2022) [113]	Adalimumab	160 mg s.c. at week 0, 80 mg at week 2, and then 40 mg every week from week 4	Open-label study	UC; RA; hypertension; hyperlipidemia; hyperuricemia; osteoporosis	12 (54.5%) of 22 patients reached a satisfactory outcome
M. Seishima et al. (2022) [114]	Adalimumab	160 and 80 mg given s.c., biweekly, and then 40 mg weekly	Case report	History of systemic sarcoidosis; renal failure	Satisfactory result
S. Ohmura et al. (2023) [115]	Adalimumab	NA	Case report	RA	Satisfactory result
A. Campanati et al. (2015) [116]	Adalimumab	160 mg s.c. at week 0, 80 mg at week 1, and then 40 mg every 2 weeks	Case report	CD	Complete healing after 12 weeks

s.c.—subcutaneously; IBD—inflammatory bowel disease; UC—ulcerative colitis; CD—Crohn’s disease; RA—rheumatoid arthritis; NA—not applicable

**Table 4 ijms-25-02440-t004:** A summary of clinical studies concerning etanercept, LE-lupus erythematosus.

Authors (Year)	Biologic Drug	Dosage Regimen	Study Type	Comorbidities	Efficacy
M. Ariane et al. (2019) [121]	Etanercept	50 mg per week s.c.	Case report	None known; breast plastic surgery	Complete remission
F. S. Kim et al. (2012) [122]	Etanercept	50 mg twice weekly; at 9 months, 50 mg per week s.c.	Case report	CD	Satisfactory result
D. B. Roy et al. (2006) [123]	Etanercept	25 mg twice weekly s.c.	Case reports (*n* = 3)	(1) RA, LE (2) RA, hypothyroidism, DVT (3) None known	Complete healing after 2 months in patients 1 and 3; satisfactory result in patient 2
F. J. Rogge et al. (2008) [124]	Etanercept	50 mg per week s.c.	Case report	None known	Complete healing after 7 months
V. Haridas et al. (2017) [125]	Etanercept	1 mg/2 × 2 cm area—topical	Case report	Sjogren’s syndrome	Satisfactory result
G. Goldenberg et al. (2005) [126]	Etanercept	25 mg twice weekly s.c.	Case report	Autoimmune hepatitis	Complete healing after 5 months
N. Pastor et al. (2005) [127]	Etanercept	25 mg twice weekly s.c.	Case report	NA	Complete healing after 8 weeks
JW 4th McGowan et al. (2004) [128]	Etanercept	NA	Case report	NA	Satisfactory result
R. Guedes et al. (2012) [129]	Etanercept	NA	Case report	NA	Satisfactory result
M. M. Kleinpenning et al. (2011) [132]	Etanercept	50 mg twice weekly s.c.	Case report	Hypogammaglobulinemia	Insufficient clinical improvement

s.c.—subcutaneously; CD—Crohn’s disease; RA—rheumatoid arthritis; DVT—deep vein thrombosis; NA—not applicable; *n*—number.

**Table 5 ijms-25-02440-t005:** A summary of clinical studies concerning ustekinumab.

Authors (Year)	Biologic Drug	Dosage Regimen	Study Type	Comorbidities	Efficacy
M. Benzaquen et al. (2017) [118]	Ustekinumab	45 mg s.c.	Case report	psoriasis	Satisfactory result
I. A. Vallerand et al. (2019) [133]	Ustekinumab	520 mg iv. Infusion at week 0, 90 mg s.c. at week 8 and then every 8 weeks	Case report	MG, DM, hypertension, dyslipidemia, CKD, gout, and obstructive sleep apnea	Complete healing after 6 months
J. López González et al. (2021) [134]	Ustekinumab	260 mg iv. Infusion, then 90 mg s.c. every 8 weeks	Case report	CD	Satisfactory result
M. Fahmy et al. (2012) [135]	Ustekinumab	90 mg s.c. at weeks 0 and 2, then every 8 weeks beginning at week 10	Case report	UC	Complete healing by week 10
Z. M. Low et al. (2018) [136]	Ustekinumab	90 mg s.c. at weeks 0 and 4, then every 6 weeks, and later 45 mg every 3 weeks	Case report	NA	Significant improvement at 3 months
P. García Cámara et al. (2019) [137]	Ustekinumab	520 mg iv. Infusion at week 0, then 90 mg s.c. every 8 weeks	Case report	CD	Complete healing after 12 months
J. Piqueras-García et al. (2019) [138]	Ustekinumab	90 mg s.c. at weeks 0, 4, 10, and every 8 weeks thereafter	Case report	UC	Satisfactory result
D. Nieto et al. (2019) [139]	Ustekinumab	90 mg s.c. every 8 weeks	Case report	Myelodysplastic syndrome	Complete healing after 20 weeks
A. M. Goldminz et al. (2012) [140]	Ustekinumab	90 mg s.c. at weeks 0 and 4, and then every 8 weeks	Case report	None known	Satisfactory results after 22 weeks
A. J. Petty et al. (2020) [141]	Ustekinumab	90 mg s.c. at weeks 0 and 4, and then every 8 weeks	Case report	Psoriasis and palmoplantar pustulosis	Satisfactory results after 2 doses
I. Cosgarea et al. (2016) [142]	Ustekinumab	NA	Case report	Renal cell carcinoma, chronic venous insufficiency, diabetes, hypertension	Complete healing after 3 months
E. Guenova et al. (2011) [143]	Ustekinumab	45 mg s.c. at week 0 and week 4	Case report	None known	Complete healing after 14 weeks
G. Nunes et al. (2019) [144]	Ustekinumab	520 mg iv. Infusion, then 90 mg s.c. every 8 weeks	Case report	CD	Satisfactory result

s.c.—subcutaneously; MG—myasthenia gravis; CKD—chronic kidney disease; DM—diabetes mellitus; CD—Crohn’s disease; UC—ulcerative colitis; NA—not applicable.

**Table 6 ijms-25-02440-t006:** A summary of clinical studies concerning IL-1 antagonists.

Authors (Year)	Biologic Drug	Dosage Regimen	Study Type	Comorbidities	Efficacy
A. G. A. Kolios et al. (2015) [145]	Canakinumab	150 mg s.c. at weeks 0 and 2, then 150–300 mg at week 4 if needed	Prospective, open-label study	none known	Complete healing in 4 out of 5 patients
S. Acierno et al. (2022) [146]	Canakinumab	4 mg/kg s.c. every 4 weeks, after a year, 4 mg/kg every 8 weeks, because of exacerbation of the disease after a year of remission, return to the dosage of 4 mg/kg every 4 weeks	Case report	refractory chronic recurrent multifocal osteomyelitis	Satisfactory response
T. Jaeger et al. (2013) [147]	Canakinumab	150 mg s.c. every 3–6 weeks, a total of 8 injections	Case report	HS	Complete remission in 1 year
C. O’Connor et al. (2021) [148]	Anakinra	2 mg/kg s.c. daily in 4 weeks, then 100 mg daily	Case report	(1) obesity, APS (2) CKD, gout, hypertension, peripheral vascular disease, and dyslipidemia	Complete healing in 4 months

s.c.—subcutaneously; APS—antiphospholipid syndrome; CKD—chronic kidney disease; HS—hidradenitis suppurativa.

**Table 7 ijms-25-02440-t007:** A summary of clinical studies concerning IL-17 antagonists.

Authors (Year)	Biologic Drug	Dosage Regimen	Study Type	Comorbidities	Efficacy
J. Coe et al. (2022) [149]	Secukinumab	300 mg s.c. four weekly; after 2 months, 300 mg two weekly	Case report	Depression, osteoarthritis, hiatus hernia, Gilbert’s syndrome, and previous hepatitis A infection	Complete healing after a year of high-dose therapy
A.S. Kao et al. (2023) [150]	Ixekizumab	160 mg s.c. at week 0, then 80 mg every 2 weeks until week 12, then 80 mg every 4 weeks	Case series	(1) HS, history of PG, SARS-CoV-2 (2) None (3) None (4) Metastatic renal cell carcinoma	(1) Complete response (2) Complete response (3) Clinical improvement (4) Near complete healing after 12 months
M. W. Tee et al. [151]	Brodalumab	210 s.c. every week	Case series	(1) Acne conglobate, HS (PASH) (2) HS	Complete healing in both cases
M.L. McPhie et al. (2020) [152]	Secukinumab	300 mg s.c. at weeks 0, 1, 2, 3, and 4, followed by monthly maintenance dosing	Case report	NA	Complete healing
M.M. Garcia et al. (2018) [153]	Secukinumab	300 mg s.c. at weeks 0, 1, 2, 3, and 4, then every 4 weeks; beginning week 16, 300 mg every other week	Case report	RA, post-surgery for Quervain’s tenosynovitis	Partial response after 20 months of treatment

HS—hidradenitis suppurativa; PG—pyoderma gangrenosum; s.c.—subcutaneously; NA—not applicable; RA—rheumatoid arthritis.

**Table 8 ijms-25-02440-t008:** A summary of clinical studies concerning IL-23 antagonists.

Authors (Year)	Biologic Drug	Dosage Regimen	Study Type	Comorbidities	Efficacy
L. J. Leow et al. (2022) [161]	Tildrakizumab	100 mg s.c. on week 0 and 4, then every 8 weeks	Case report	NA	Constant improvement after 82 weeks
E. Çalışkan et al. (2023) [163]	Risankizumab	NA	Case report	ankylosing spondylitis, ileostomy due to megacolon toxicum	Refractory to treatment; closed primary ostomy—regression of lesions; no new lesions at the side of new ostomy
C. Baier et al. (2020) [164]	Guselkumab	100 mg s.c. monthly	Case report	monoclonal gammopathy of undetermined significance and type 2 diabetes	Complete healing within 3 months
A. M. Reese et al. (2022) [165]	Guselkumab	200 mg s.c. at week 0, 100 mg at week 4, then every 6 weeks	Case report	type 2 diabetes mellitus	Complete healing after 4 doses
J. M. John et al. (2020) [162]	Tildrakizumab	100 mg s.c. on week 0, 4, then every 12 weeks	Case report	gout, polymyalgia rheumatica, renal impairment	Almost complete healing
B. Burgdorf et al. (2020) [166]	Risankizumab	150 mg s.c. on weeks 0, 4, then every 12 weeks	Case report	none	Significant improvement
L.V. Piñeiro et al. (2023) [167]	Guselkumab	100 mg s.c. at week 0, 4, then every 8 weeks	Case report	NA	Complete healing with residual post-inflammatory lesions

NA—not applicable; s.c.—subcutaneously.

## Data Availability

Not applicable.

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
