# Peer review of "The Pathophysiology and Treatment of Pyoderma Gangrenosum—Current Options and New Perspectives"

_ijms, 2024, doi:10.3390/ijms25042440_

Round 1
Reviewer 1 Report
Comments and Suggestions for Authors
Comments on the Quality of English LanguageAuthor Response
Dear Reviewer,
We sincerely thank you for your thorough evaluation of our review on pyoderma gangrenosum. Please see the attachment.

Reviewer 2 Report
Comments and Suggestions for Authors
The authors present a review focused on pathophysiology and treatments of Pyoderma gangrenosum. The manuscript is well-constructed but, some areas need to be improved. In particular, "Pathogenesis" section doesn't report some recent important research in this field. The authors cited some "old" research, but they should take these into consideration as well:
1) Moura RR, Brandão L, Moltrasio C, Agrelli A, Tricarico PM, Maronese CA, Crovella S, Marzano AV. Different molecular pathways are disrupted in Pyoderma gangrenosum patients and are associated with the severity of the disease. Sci Rep. 2023 Mar 25;13(1):4919. doi: 10.1038/s41598-023-31914-z.
Concerning syndromic PG (line 105-109)
1) Genovese G, Moltrasio C, Garcovich S, Marzano AV. PAPA spectrum disorders. G Ital Dermatol Venereol. 2020 Oct;155(5):542-550. doi: 10.23736/S0392-0488.20.06629-8.
2) Marzano AV, Genovese G, Moltrasio C, Tricarico PM, Gratton R, Piaserico S, Garcovich S, Boniotto M, Brandão L, Moura R, Crovella S. Whole-Exome Sequencing in 10 Unrelated Patients with Syndromic Hidradenitis Suppurativa: A Preliminary Step for a Genotype-Phenotype Correlation. Dermatology. 2022;238(5):860-869. doi: 10.1159/000521263.
A figure illustrating the main molecular pathways and related targeted treatments would be helpful to readers.
Comments on the Quality of English LanguageMinor editing of English language are required.
Author Response
Dear Reviewer,
We appreciate your thoughtful review of our manuscript on the pathophysiology and treatment of pyoderma gangrenosum. Please see the attachment.

Reviewer 3 Report
Comments and Suggestions for Authors
The authors reviewed pathogenesis, current treatments, and potential new treatment options for pyoderma gangrenosum(PG). The narrative review is well-written and has clinical significance. I have several suggestions for the authors.
1. In Introduction section, the authors cite reference 8 and state "Six clinical variants of PG can be distinguished[8]". The reference described that there are 4 types of PG (ulcerative, bullous, pustular, and vegetative). Also, the authors didn't mention pustular type in Introduction.
2. Reference 10 described the incidence of PG was 0.63 per 100,000 person-years. Incidence is different from prevalence.
3. For biologic agents, there are also reports that PG could be treated by IL-17 inhibitors (DOI: 10.1111/dth.15669, DOI: 10.1016/j.jdcr.2023.05.002, DOI: 10.1111/dth.15716; DOI: 10.1111/1346-8138.17031) and IL-23 inhibitors (DOI: 10.1111/1346-8138.17031, DOI: 10.1111/jdv.19555, DOI: 10.2147/CCID.S374534). As the authors described IL-23-IL-17 axis in the pathogenesis of PG, the authors may discuss the role of IL-23 inhibitors and IL-17 inhibitors for treating PG to make this review article more comprehensive.
4. The content in Conclusions section is too general. As the authors focus on treatments, it is suggested to summarize important or potential new treatment options in Conclusions.
Author Response
Dear Reviewer,
We are very thankful for your precious time spent reviewing our manuscript and all your valuable comments and suggestions. Please see the attachment.

Round 2
Reviewer 2 Report
Comments and Suggestions for Authors
The manuscript can be accepted in the current form.
Reviewer 3 Report
Comments and Suggestions for Authors
The authors have substantially revised the manuscript in respose to previous reviewers' comments. I have no more suggestions.